# Production Decisions of Automakers Considering the Impact of Anticipated Regret under the Dual-Credit Policy

**Yushi Wang, Licheng Sun * and Shilong Li**

School of Management, Jiangsu University, Zhenjiang 212013, China; wang807475252@163.com (Y.W.); 1000003731@ujs.edu.cn (S.L.)

* Correspondence: sunsee213@ujs.edu.cn; Tel.: +86-158-5045-5868

**Abstract:** The anticipation of consumer regret under dual-credit policies significantly impacts automakers' production decisions. This article focuses on the automakers that produce both new energy vehicles (NEV) and fuel vehicles (FV). Given the dual-credit policy, this study introduces the concept of anticipated regret to characterize consumers' evaluation of product utility, and then analyzes the impact of this behavior on the volume of vehicles produced by automakers. The study found the following: when in independent decision-making mode, as the government increases the requirements associated with the new energy vehicle credit ratio, automakers reduce the number of fuel vehicles and the number of new energy vehicles produced. In this independent decision-making mode, the influence of consumer anticipation of regret on automakers' production decisions is uncertain. When the credit price is less than the threshold, the total profit of the automaker in a centralized decision-making mode is less than the profit in the independent decision-making mode. When the credit price exceeds the threshold, the total profit of the automaker is greater in the centralized decision-making mode compared with the independent decision-making mode.

**Keywords:** new energy vehicle; dual-credit policy; anticipated regret; production decision

## 1. Introduction

As the pressure to decrease emissions increases, many scholars and governments have focused on new energy vehicles (NEV), given their excellent environmental effects [1–4]. To encourage the development of new energy vehicles and optimize the production decisions of automakers producing new energy vehicles, the Chinese government began to implement a subsidy policy in 2010. However, the subsidy policy lacks oversight and has a significant crowding effect with respect to the research and development of new energy vehicles [5,6].

To this end, in 2018, the Chinese government officially promulgated and implemented Parallel Management Measures for Average Fuel Consumption of Passenger Vehicles and New Energy Vehicle Credits (hereinafter referred to as the "dual-credit policy"). Although this policy has somewhat optimized the production decisions of Chinese automakers and the transformation of the automobile industry [7,8], if the dual-credit policy is too biased towards new energy vehicles, fuel cars will be pushed out of the market too quickly. This could increase the instability of the automobile market, and further increase the difficulty facing automakers when making production decisions [9–11].

At the same time, from the perspective of consumer purchasing decisions, product attributes, such as endurance and functional quality (such as battery reliability), significantly impact the intention of consumers to purchase new energy vehicles [12–14]. When consumers do not fully understand the attributes of automobile products, they may anticipate regret before making the purchase decision; this anticipated regret may also affect the production decisions of automakers. For example, Tesla has introduced a seven-day no-reason return plan to help mitigate the risk of anticipated regret for consumers.

The dual-credit policy incentivizes automakers to produce new energy vehicles. However, the anxiety associated with long trips and safety problems associated with the batteries in new energy vehicles may lead consumers to anticipate regretting their purchase of new energy vehicles. This may interfere with consumers' purchasing decisions, which in turn affects the production decisions of automakers. Therefore, under the dual-credit policy, this paper introduces the psychology of consumers' anticipated regret, analyzes the production decisions of automakers, and explores the optimal production decisions.

## 2. Literature Review

The research related to this paper focuses on two aspects: production decisions and the anticipated regret of consumers.

### 2.1. Production Decisions of Automakers

Scholars have studied the production decision-making problem of automakers without and with the dual-credit policy. In the absence of the dual-credit policy, scholars have focused on the impact of subsidy policies on the production decisions of automakers. Liu et al., Liu et al., and Hu et al. argued that both consumer subsidies and research and development (R&D) subsidies can affect whether an automaker develops new energy vehicles [15–17]. However, Zhang et al. found that the technological capability of an automaker has a more significant impact on its production decisions [18]. Zhu et al. posited that subsidy policies can stimulate consumers' demand for new energy vehicles [19]. However, Wang et al. and Zhu et al. pointed out that a decline in subsidies weakens consumers' enthusiasm for buying new energy vehicles [20,21]. Liu et al. found that higher R&D subsidies have a crowding out effect on the R&D of new energy vehicles [22]. Wu et al. found that the subsidy policy only effectively stimulated the production of new energy vehicles in the absence of competition [23]. Compared with the subsidy, Guo et al. hypothesized that encouraging technological progress led automakers to make sustainable production decisions [24].

In terms of the dual-credit policy, Ou et al. found that the dual-credit policy guided automakers to produce new energy vehicles [25]. The implementation of the dual-credit policy promoted cooperation between traditional automakers and new energy automakers; both sides were incentivized to reach a unified production strategy [26,27]. The "subsidy retreat" reduces the optimal yield of new energy vehicles; however, under the dual-credit policy, the optimal yield of new energy vehicles increases with an increase in scientific research input [28]. The credit ratio in the dual-credit policy cannot effectively guide the production decisions of automakers [29]. If the dual-credit policy is excessively tilted to new energy vehicles, fuel cars may be crowded out of the market too quickly, increasing the instability of the automobile market [30,31]. In addition, Li et al. constructed a GERT network model of industrial value flow to optimize NEV credit allocation [32]. They analyzed the value flow and value-added process in the field of new energy vehicles under the dual-credit policy. Guo et al. explored the effectiveness of the dual-credit policy from the perspective of cooperation between domestic and foreign automakers [33]. They proposed establishing a new energy vehicle technology standard system to prevent "malformed grafting".

### 2.2. Consumer Anticipated Regret

The psychology of anticipated regret is mainly manifested in the consumer decision-making process [34–36]. Scholars have introduced the psychology of anticipated regret into purchasing decision-making, product production, and other fields. In terms of consumer purchasing decisions, Yuen et al. and Gabillon et al. reported that anticipated regret may aggravate consumer perceptions of scarcity, and at the same time, lead consumers to avoid information when making decisions [37,38]. This, in turn, affects consumer purchase behavior. Sarangee et al. pointed out that the abandonment utility associated with anticipation of regret has a larger effect on consumers than choice utility [39].

Chen et al. introduced the anticipation of regret in consumers buying obviously counterfeit products [40]. They hypothesized that only consumers who have an independent self-view and do not care about their social faces will experience regret effect expectations. In terms of product production, Gao et al. and Feng et al. introduced anticipated regret into the remanufacturing supply chain [41–43]. The study found that anticipated regret benefits the level of technological innovation among remanufacturers when there is an unclear level of heterogeneity among consumers, improving supply chain profits. A higher level of sensitivity to expected regret can partially alleviate competition between new and remanufactured products. Liu et al. and Jiang et al. explored the impact of anticipated regret on product replacement strategies and new product development strategies, respectively [44,45]. Khan et al. pointed out that the level of consumers' economic hostility increases their anticipated regret when purchasing foreign products from hostile target markets [46]. Mourali et al. found that automakers can influence the decision-making of regret-anticipating consumers by affecting the salience of regret, the reversibility of outcomes (such as return policies), and the trajectory of regret (postpone versus choosing now) [47].

In summary, scholars have actively explored production decision-making with respect to new energy vehicles. However, there are three key research gaps.

First, scholars have not yet considered consumers' anticipated regret when purchasing a car. Extensive literature has shown that anticipated regret can significantly impact consumer decision-making [48–51]. When buying a car, it is difficult for consumers to estimate the true utility of the product. Consumers may experience anticipatory regret, impacting the production decisions of automakers. Second, few studies have explored the impact of policy and consumer psychology on the production decisions of automakers from both supply and demand perspectives. Implementing the dual-credit policy may guide the decision-making optimization and industrial transformation of automakers, but the higher sales price of new energy vehicles and their corresponding product attributes may make it difficult to impress consumers. In particular, when consumers anticipate regret, purchasing decisions are uncertain, and it is more difficult for automakers to make production decisions. Third, most previous studies have analyzed the direct impact of the dual-credit policy on production decisions. Few scholars have explored the impact of changes in the double-point policy on the points trading market. The cost of income or expenditure obtained in the credits trading market is an important part of corporate profits, so changes in the credits trading market lead to changes in production decisions.

Based on this, this paper introduces the psychology of consumers' anticipation of regret under the dual-credit policy, and explores the influence of the dual-credit policy on the production decision-making of automakers. First, a consumer surplus model is constructed according to the anticipated regret utility formula, and the influence of anticipated regret on consumer decision-making and car company production decision is analyzed. Second, the study considers the dual-point policy and the anticipated regret of consumers from both supply and demand perspectives, and explores the production decision-making issues of automakers under the joint action of both supply and demand. Finally, the study compares the impact of policy changes on automakers' credit transactions under centralized decision-making and under independent decision-making, and explores the mechanism involved in how the dual-credit policy impacts automakers' production decisions.

## 3. Methodology

To highlight the novelty more clearly, Table 1 in this section presents the main differences between the content of this paper and related research. Since the implementation of the dual-credit policy, the production decision-making of automakers has been the focus of scholars' research. Yu et al. explored the impact of the subsidy policy and the dual-credit policy on the production decisions of automakers when the two types of vehicles (NEV and FV) were in independent markets [28]. For consumers, NEV and FV, however, are substitute products, so the relationship between NEV and FV tends to be more competitive

in practice. Although Li et al. explored the optimal production decisions of automakers in a competitive market environment [8], they did not consider the psychological characteristics of consumers in a competitive environment. Tang et al. considered the impact of consumers' low-carbon preference on the production decisions of the automakers [10], but low-carbon preference only showed consumers' attention to the advantages of NEV, while ignoring consumers' attention to the advantages of FV, such as price advantage. The anticipated regret highlights the entanglement of consumers facing two competing products before making a purchase decision. In the introduction, this article explains why anticipation of regret precedes the decision to purchase cars. Therefore, this paper will investigate the influence of consumers' anticipated regret on the optimal decisions of production for the automaker under the dual-credit policy.

**Table 1.** Main differences between our works and existing research.

| Articles | Research Object | Market Environment | Focus Point | Consumer Psychology |
|---|---|---|---|---|
| Lou et al. [7] | ICEV | independent | R&D, Production | low carbon preference |
| Tang et al. [10] | NEV, FV | competitive | Price, Production | low carbon preference |
| Lu et al. [29] | NEV, FV | competitive | Price, Production, R&D | low carbon preference |
| Li et al. [9] | NEV, FV | independent | Profit | × |
| Yang et al. [52] | NEV, FV | competitive | Price Production, Profit | × |
| Li et al. [8] | NEV, FV | competitive | Production | × |
| Ou et al. [25] | NEV, FV | competitive | Production, R&D | × |
| Yu et al. [28] | NEV, FV | independent | Production, R&D | × |
| Cheng et al. [26] | NEV, FV | independent | Production | × |
| Ma et al. [53] | FV | independent | Production, R&D | × |

Regarding the method and contribution, the duopoly game model is adopted in this paper. Although this article takes one automaker as the research object, the automaker has two departments (NEV production department and FV production department, details in the problem description). Then this paper explores the influence of dual-credit policy and anticipated regret on production decision under centralized and independent decision-making. The contribution of this paper to the domain of NEV is the construction of the demand function based on the anticipated regret function. In previous papers, scholars have constructed demand functions based either on consumer surplus or on market demand. In this paper, the anticipated regret function is used to construct the demand function, which enriches the construction method of the demand function in the NEV automotive field. In the existing literature, few scholars have introduced regret anticipation psychology in the field of NEV. This paper aims to fill this gap.

The chapters of this paper are described as follows: the problems and the model assumptions are described in Section 4. Section 5 mainly conducts a theoretical analysis on the influence of dual-credit policy and the anticipated regret psychology on the production decision of the automaker. Section 6 verifies the theoretical conclusions obtained in Section 5 through numerical simulations, and attempts to find conclusions that are difficult to obtain through mathematical models Section 7 summarizes the results obtained in this paper and introduces the directions of author's future research.

## 4. Problem Statements and Model Assumptions

### 4.1. Problem Statements

With the dual-credit policy, assume there is an automaker producer in the market that produces both fuel vehicles and new energy vehicles. The amount of average fuel consumption (CAFC) credits generated by an automaker producing a fuel vehicle is $\lambda_F Q_F$. The amount of new energy vehicle (NEV) credits generated by producing new energy vehicles is $\lambda_N Q_N$. In these expressions, $Q_F$ and $Q_N$ represent the output of fuel vehicles and the output of new energy vehicles, respectively. The parameter $\lambda_F$ is the credit coefficient for producing fuel vehicles, and $\lambda_N$ is the credit coefficient for producing new energy

vehicles. The government's given NEV credits ratio requirement is $\beta$, and the NEV credit reaches the standard value of $\beta Q_F$. The tradable part of the NEV positive credits generated from producing new energy vehicles is $\lambda_N Q_N - \beta Q_F$. From a demand perspective, since consumers are affected by anticipated regret, they consider the negative effects of current decisions before purchasing a car. For example, factors such as anxiety about mileage are considered before purchasing a new energy vehicle; factors such as travel restrictions are considered before purchasing a fuel vehicle. These factors may impact consumer behavior and automakers' production decisions.

### 4.2. Basic Assumptions

To explore the influence of dual-credit policy and anticipated regret on automakers' production decisions, this paper makes the following assumptions:

**Assumption 1.** *Let the willingness of consumers to pay for new energy vehicles be "$v$, $v \in [0, 1]$" and follow a uniform distribution within the range of $[0, 1]$. Before the purchase, consumers do not know which product they prefer, since they do not know enough about the attributes of the two types of products. Attributes include usage rights, environmental protection, and maintenance costs. To simplify the analysis, there are Type $\theta_H$ and Type $\theta_L$ consumers $(0 < \theta_L < \theta_H < 1)$. Type $\theta_H$ consumers highly prefer fuel vehicles; Type $\theta_L$ consumers have a low preference for fuel vehicles, and the market shares are equal for Type $\theta_H$ and Type $\theta_L$ consumers.*

**Assumption 2.** *Let $P_N$ and $P_F$ be the sales prices of new energy vehicles and fuel vehicles, respectively. The consumer surplus of Type $\theta_L$ consumers buying fuel vehicles is $\theta_L v - P_F$. The consumer surplus of Type $\theta_H$ consumers buying fuel vehicles is $\theta_H v - P_F$. That is, Type $\theta_H$ consumers are incentivized to buy fuel vehicles than Type $\theta_L$ consumers. Type $\theta_L$ consumers are more incentivized to buy new energy vehicles than Type $\theta_H$ consumers. Consumers do not know which type they belong to before buying. They only know that the probability that they belong to Type $\theta_H$ and Type $\theta_L$ is equal, so the consumer's expected preference for fuel vehicles is $\frac{\theta_L + \theta_H}{2}$. The expected utility of consumers buying fuel vehicles is $\frac{\theta_L + \theta_H}{2} - P_F$, and the expected utility of purchasing new energy vehicles is $v - P_N$.*

**Assumption 3.** *After purchasing and using a vehicle, consumers understand their actual type, which may result in the experience of regret. After experiencing regret many times, consumers may anticipate regret before purchasing. Anticipated regret is a feature of risk aversion psychology, and is reflected in the negative effect of the product purchase process.*

**Assumption 4.** *Assume that all consumers have anticipation of regret. When consumers choose to buy new energy vehicles, Type $\theta_H$ consumers experience future regret, with a 0.5 probability. In the same way, when consumers choose to buy fuel vehicles, Type $\theta_L$ consumers experience future regret, with a 0.5 probability.*

**Assumption 5.** *Assume an automaker produces a single new energy vehicle model and a single fuel vehicle model. Producing new energy vehicles generates NEV positive credits, and producing fuel vehicles leads to negative CAFC credits. This paper focuses on the production decisions of automakers in a single cycle. As such, the negative CAFC credits need to be offset by purchasing NEV positive credits. The offset ratio is 1:1; NEV positive points must enter the points market for trading.*

Table 2 lists the variables considered in this paper.

Income from producing fuel vehicles, credit price, NEV point ratio requirements, consumer surplus for purchasing new energy vehicles, consumer surplus for purchasing fuel vehicles, consumer preference coefficient for fuel vehicles.

**Table 2.** Symbols and descriptions of parameters and variables.

| Symbol | Meaning |
|---|---|
| $\lambda_N$ | Unit NEV credit coefficient |
| $\lambda_F$ | Unit CAFC credit coefficient |
| $P_N$ | New energy vehicle selling price |
| $P_F$ | Fuel vehicle selling price |
| $Q_N$ | New energy vehicle production |
| $Q_F$ | Fuel vehicle production |
| $\pi_N$ | Income from producing new energy vehicle |
| $\pi_F$ | Income from producing fuel vehicles |
| $P_\lambda$ | Credit price |
| $\beta$ | NEV credit ratio requirements |
| $U_N$ | Consumer surplus for purchasing new energy vehicles |
| $U_F$ | Consumer surplus for purchasing fuel vehicles |
| $\theta$ | Consumer preference coefficient for fuel vehicles |

## 5. Model Building and Analysis

Jiang's research denotes the anticipated regret disutility as $A.R. = -\gamma_i \times prob(U_f > U_c) \times \left(U_f - U_c\right)$ [45]. In this expression, $U_f$ is the net utility of the consumer's decision to give up; $U_c$ is the net utility of the consumer's choice decision; and $prob(U_f > U_c)$ represents the probability that the utility of the decision to give up is greater than the utility of the decision to choose, or, the probability of regret. The expression $\gamma_i[i \in (f,c)]$ represents the regret sensitivity coefficient, which indicates the sensitivity of consumers to the expectation of giving up and choosing regret. This paper establishes $\gamma_f = \gamma_c = \gamma$ [10,11].

When Type $\theta_H$ consumers experience regret, $U_c = v - P_N$ and $U_f = \theta_H v - P_F$. Then, the anticipated disutility of regret is expressed as $U_n = -\frac{\gamma}{2}[(\theta_H v - P_F) - (v - P_N)]$. When Type $\theta_L$ consumers experience regret, $U_c = \theta_L v - P_F$ and $U_f = v - P_N$. Then, the anticipated disutility of regret is expressed as $U_{fv} = -\frac{\gamma}{2}[(v - P_N) - (\theta_L v - P_F)]$.

Given the above, the expected net utility values associated with consumers buying new energy vehicles and fuel vehicles are expressed in Equations (1) and (2), respectively:

$$U_N = v - P_N - \frac{\gamma}{2}[(\theta_H v - P_F) - (v - P_N)] \tag{1}$$

$$U_F = \frac{\theta_L + \theta_H}{2}v - P_F - \frac{\gamma}{2}[(v - P_N) - (\theta_L v - P_F)] \tag{2}$$

Based on the expressions $U_N = 0$, $U_N = U_F$, and $U_F = 0$, respectively, the payoff critical points are calculated as:

$$v_1 = \frac{2P_N - \gamma(P_F - P_N)}{2 + \gamma(1 - \theta_H)}, \quad v_2 = \frac{2(P_F - P_N)}{\theta_L + \theta_H - 2}, \quad v_3 = \frac{P_F(2 + \gamma) - \gamma P_N}{\theta_H + \theta_L(1 + \gamma) - \gamma}$$

when $v > v_2$, consumers buy new energy vehicles. When $v_2 > v > v_3$, consumers buy fuel vehicles. From this, the production quantities of the two products are calculated as:

$$Q_N = \int_{v_2}^{1} f(v)dv = 1 - \frac{2(P_F - P_G)}{2 - \theta_H - \theta_L} \tag{3}$$

$$Q_F = \int_{v_3}^{v_2} f(v)dv = \frac{2(P_F - P_N)}{\theta_H + \theta_L - 2} - \frac{P_F(2 + \gamma) - \gamma P_N}{\theta_H + \theta_L(1 + \gamma) - \gamma} \tag{4}$$

Assume that $\theta = \frac{\theta_H + \theta_L}{2}$ represents the expected acceptance of fuel vehicles by consumers; assume that $k = \frac{\theta_H}{\theta_L}$ represents the difference in the acceptance of two types of consumers for fuel vehicles. This paper focuses on exploring the impact of the double-point policy and consumer anticipation of regret on the production decisions of automakers. As such, the actual difference in consumer acceptance is not the main research topic, and $k = 2$.

For convenience, assume $6\theta - \gamma\theta$ is A, $6 - \gamma\theta$ is B, $-3\gamma + 6\theta + 2\gamma\theta$ is F, and $1 - \theta$ is G. The relationship is expressed as: $0 < G < F < A < B$. The demand for the two products is expressed as:

$$Q_N = 1 - \frac{P_N - P_F}{G} \tag{5}$$

$$Q_F = \frac{A}{GF}P_N - \frac{B}{GF}P_F \tag{6}$$

*5.1. The Independent Decision-Making Model of Two Production Departments of an Automaker*

In the independent decision-making mode, two production departments associated with an automaker make decisions in ways that maximize their own interests. The negative CAFC credits needed for the fuel vehicle production department are bought from the credits trading market. The NEV positive credits of the new energy vehicle department are sold in the credits market. The profit generated by the two departments is:

$$\pi_F = (P_F - C_F)Q_F - P_\lambda(\lambda_F Q_F + \beta Q_F) \tag{7}$$

$$\pi_N = (P_N - C_N)Q_N + P_\lambda \lambda_N Q_N \tag{8}$$

These expressions are solved as follows:

$$P_F^* = -\frac{2BC_F + AC_F + AG + 2BP_\lambda\beta + 2BP_\lambda\lambda_F - AP_\lambda\lambda_N}{A - 4B} \tag{9}$$

$$P_N^* = -\frac{2B(C_F + 2C_N + 2G + P_\lambda\beta + P_\lambda\lambda_F - 2P_\lambda\lambda_N)}{A - 4B} \tag{10}$$

Placing the optimal sales price into Equations (5) and (6), the optimal output of fuel vehicles and new energy vehicles is expressed as:

$$Q_F^* = -\frac{B(-2B(C_F + P_\lambda(\beta + \lambda_F)) + A(C_F + C_N + G + (\beta + \lambda_F - \lambda_N)))}{FG(A - 4B)} \tag{11}$$

$$Q_N^* = \frac{-AC_N + AP_\lambda\lambda_N - B(C_F - 2C_N + 2G + P_\lambda(\beta + \lambda_F + 2\lambda_N))}{G(A - 4B)} \tag{12}$$

Substituting Equations (9)–(12) into Equations (7) and (8), respectively, the maximum profit of the two types of cars is expressed as:

$$\pi_F^* = \frac{B(-2B(C_F + P_\lambda(\beta + \lambda_F)) + A(C_F + C_N + G + (\beta + \lambda_G - \lambda_N)))^2}{FG(A - 4B)^2} \tag{13}$$

$$\pi_N^* = \frac{(A(CN - P_\lambda\lambda_N) + B(C_F - 2C_N + 2G + P_\lambda(\beta + \lambda_F + 2\lambda_N)))^2}{G(A - 4B)^2} \tag{14}$$

The solution reflects the optimal production quantities of new energy vehicles and fuel vehicles under independent decision-making, expressed as $(Q_N^*, Q_F^*)$, respectively. The optimal price of new energy vehicles is $P_N^*$, and the optimal price of fuel vehicles is $P_F^*$. The profit is $\pi_N^*$, and the profit generated by fuel vehicles is $\pi_F^*$.

**Proposition 1.** *The production quantity $Q_N$ of new energy vehicles increases as the unit NEV credit coefficient $\lambda_N$ increases. This upward trend is strengthened as the regret sensitivity coefficient $\gamma$ increases. The production quantity $Q_F$ of fuel vehicles decreases as the unit NEV credit coefficient $\lambda_N$ increases. This downward trend strengthens as the regret sensitivity coefficient $\gamma$ increases.*

**Proof 1.**

$$\frac{\partial Q_N^*}{\partial \lambda_N} = \frac{P_\lambda(A - 2B)}{G(A - 4B)} > 0, \quad \frac{\partial^2 Q_N^*}{\partial \lambda_N \partial \gamma} = \frac{4\theta P_\lambda}{3(-8 + \theta(2 + \gamma))^2} > 0$$

$$\frac{\partial Q_F^*}{\partial \lambda_N} = \frac{ABP_\lambda}{AFG - 4BFG} < 0,$$

$$\frac{\partial^2 Q_F^*}{\partial \lambda_N \partial \gamma} = \frac{2\theta P_\lambda \left(-432 + \theta\left(\gamma^2(3 - 11\theta) - 12\gamma(\theta - 9) + 36(3 + \theta)\right)\right)}{3(-8 + \theta(2 + \gamma))^2} < 0$$

Proposition 1 shows that: as the NEV credit coefficient $\lambda_N$ increases, the NEV positive credits obtained by the production unit of new energy vehicles increase. This enables automakers to obtain higher returns in the credit transaction, reducing the production cost and subsequent sales prices of new energy vehicles. This reduction in the sales price of new energy vehicles alleviates the expected regret of consumers when purchasing new energy vehicles, and increases the expected regret when purchasing fuel vehicles. Therefore, when the future sensitivity coefficient $\gamma$ gradually increases, consumers prefer to buy new energy vehicles. This encouragement to purchase new energy vehicles further encourages automakers to produce these vehicles. Therefore, the government should properly adjust the NEV credit coefficient to ensure the stability of the automobile market and encourage the development of the new energy automobile industry. The current policy only addresses the battery life of new energy vehicles. The government should also assess new energy vehicles from multiple dimensions and guide consumers to view new energy vehicles in a correct and comprehensive manner. □

**Proposition 2.** *The production of new energy vehicles increases as the credit coefficient of unit fuel vehicles increases. This upward trend weakens as the regret sensitivity coefficient increases. The production volume of fuel vehicles decreases as the unit fuel vehicle credit coefficient increases. This decreasing trend increases as the regret sensitivity coefficient increases.*

**Proof 2.**

$$\frac{\partial Q_N^*}{\partial \lambda_F} = -\frac{BP_\lambda}{G(A - 4B)} > 0, \quad \frac{\partial^2 Q_N^*}{\partial \lambda_F \partial \gamma} = -\frac{2\theta P_\lambda}{3(-8 + \theta(2 + \gamma))^2} < 0$$

$$\frac{\partial Q_F^*}{\partial \lambda_F} = -\frac{BP_\lambda(A - 2B)}{AFG - 4BFG} < 0,$$

$$\frac{\partial^2 Q_F^*}{\partial \lambda_F \partial \gamma} = \frac{2P_\lambda \left(-864 + \theta\left(\gamma^2\theta(-15 + \theta) - 12\gamma(\theta - 3)(6 + \theta) + 36\left(10 + \theta - \theta^2\right)\right)\right)}{3(-8 + \theta(2 + \gamma))^2(3\gamma - 2\theta(3 + \gamma))^2} < 0$$

Proposition 2 shows that as the CAFC credit coefficient increases, the negative CAFC credits created by producing fuel vehicle increases. This increases the production cost of fuel vehicles, increases the sales price of fuel vehicles, and increases consumer purchases of fuel vehicles anticipated regret anxiety. However, the sales price of fuel vehicles after the increase remains lower compared to new energy vehicles, and the increase in the CAFC credit coefficient does not significantly alleviate consumer anxiety about expected regret when purchasing new energy vehicles. Therefore, when the CAFC credit coefficient increases, as the regret sensitivity coefficient increases, the production of new energy vehicles shows an upward trend. However, the upward trend gradually slows down. □

To maintain market stability, the government should appropriately adjust the intensity of the dual-credit policy. If the intensity is too low, the dual-credit policy does not significantly impact the development of new energy vehicles. If the intensity is too high, new energy vehicles will overcome the production of fuel vehicles. From the perspective of improving consumers' purchasing utility, the government should encourage automakers to develop and make breakthroughs in new energy bottleneck technologies to improve the overall performance of their products. Fuel vehicle automakers can also reduce expected regret by reducing average fuel consumption and improving the negative utility of fuel conversion rates.

**Proposition 3.** *As the NEV credit ratio requirement increases, the production of new energy vehicles increases, the production of fuel vehicles decreases, and the price of both types of vehicles increases. The price of fuel vehicles increases more than the price of new energy vehicles.*

**Proof 3.**

$$\frac{\partial Q_N^*}{\partial \beta} = -\frac{P_\lambda B}{AG - 4BG} > 0, \ \frac{\partial P_N^*}{\partial \beta} = -\frac{BP_\lambda}{A - 4B} > 0$$

$$\frac{\partial Q_F^*}{\partial \beta} = -\frac{BP_\lambda (A - 2B)BP_\lambda}{FG(A - 4B)} < 0, \ \frac{\partial P_F^*}{\partial \beta} = -\frac{2BP_\lambda}{A - 4B} > 0$$

and

$$0 > \frac{\partial P_N^*}{\partial \beta} > \frac{\partial P_F^*}{\partial \beta}$$

Proposition 3 shows that as the NEV credit ratio requirement increases, the positive NEV credit required to produce a unit quantity of fuel vehicles increases. This, in turn, increases the production cost of fuel vehicles, leading to an increase in the sales price of fuel vehicles. The increase in price increases the anticipation of regret when purchasing fuel vehicles, leading some consumers to turn to new energy vehicles. When the production volume of new energy vehicles rises, the new energy vehicle production sector increases profits by raising the sales price; however, since consumers have some expected level of regret, the increased price is lower compared to fuel vehicles. □

**Proposition 4.** *As the price of credits increases, the expected negative utility associated with consumer regret for purchasing new energy vehicles gradually decreases. (That is, the expected negative utility of consumers' regret for purchasing new energy vehicles gradually weakens). The expected negative utility of regret associated with purchasing fuel vehicles gradually increases. (That is, consumers' expected regret from purchasing new energy vehicles gradually increase). The production of new energy vehicles increases, and the production of fuel vehicles declines.*

**Proof 4.**

$$\frac{\partial U_n}{\partial P_\lambda} = -\frac{\partial U_{fv}}{\partial P_\lambda} = \frac{\gamma(-6 + \gamma\theta)(\beta + \lambda_F) + \gamma(-12 + (6 + \gamma)\theta)\lambda_N}{-48 + 6\theta(2 + \gamma)} > 0$$

$$\frac{\partial Q_N^*}{\partial P_\lambda} = \frac{A\lambda_N - B(\beta + \lambda_F + 2\lambda_N)}{G(A - 4B)} > 0$$

$$\frac{\partial Q_F^*}{\partial P_\lambda} = \frac{B(-A + 2B)(\beta + \lambda_F) + AB\lambda_N}{GF(A - 4B)} < 0$$

**Proposition 5.** *There is a credit price* $P_\lambda^{(1)} = \frac{-2BC_F + A(C_F + C_N + G)}{-(A - 2B)(\beta + \lambda_F) + A\lambda_N}$, *when* $0 < P_\lambda < P_\lambda^{(1)}$. *The income of fuel vehicles decreases as the credit price increases. When* $P_\lambda^{(1)} < P_\lambda$, *the income of fuel vehicles increases with the increase in credit price.*

**Proof 5.** $\frac{\partial^2 \pi_F^*}{\partial P_\lambda^2} = \frac{2B((A - 2B)(\beta + \lambda_F) - A\lambda_N)^2}{FG(A - 4B)^2} > 0$, *when* $0 < P_\lambda < P_\lambda^{(1)}$, $\frac{\partial \pi_N^*}{\partial P_\lambda} < 0$, *when* $P_\lambda^{(1)} < P_\lambda$, $\frac{\partial \pi_F^*}{\partial P_\lambda} > 0$.

Proposition 5 shows that when the credit price is lower than the threshold, as the credit price increases, the total revenue brought by the production of fuel vehicles decreases. When the credit price exceeds the threshold, as the credit price increases, producing fuel vehicles leads to increases in total revenue. The negative credits generated by producing fuel vehicles need to be offset by NEV positive credits. As the price of credits rises, the offset cost rises, resulting in an increase in the selling price and a decrease in sales. This results in a decrease in the revenue from fuel vehicles. When the price of credits is high enough,

the cost of offsetting negative CAFC credits becomes a major component of production costs. Therefore, when the price of credits continues to rise, there is an increase in the unit selling price of fuel vehicles and a decrease in sales volume. The total negative CAFC credits generated decrease, offsetting the reduction in costs, and increasing the total revenue created by fuel vehicles. To generate fewer negative CAFC credits, automakers can reduce the average fuel consumption of fuel-powered vehicles, thereby reducing negative credits to offset costs. □

From the government's perspective, new energy vehicles have many problems at this stage, including poor battery life, and high prices for the same performance. Purchasing fuel vehicles remains more useful for consumers at this time. Therefore, at this stage, the government should adjust the price of points in a timely manner to promote the development of new energy vehicles. At the same time, the government needs to prevent fuel vehicles from being squeezed out of the market. Further, the government should assess new energy vehicles from multiple dimensions to improve their overall attributes.

### 5.2. A Centralized Decision-Making Model for Two Production Departments of an Automaker

Under centralized decision-making, the two production departments of the automaker make decisions to maximize the overall interests of the enterprise. Positive NEV credits are generated by producing the two types of cars; the required ratio of positive NEV credits and negative CAFC credits are first offset between the two production departments. Then, the remaining credits are brought into or pulled from the credits trading market. The total profit of the automaker is expressed as:

$$\pi = (P_F - C_F)Q_F + (P_N - C_N)Q_N + P_\lambda(\lambda_N Q_N - \beta Q_F - \lambda_F Q_F) \tag{15}$$

Placing $P_N$ and $P_F$ in Equation (9) generates optimal pricing for the two types of cars:

$$P_N^{**} = \frac{A^2(C_F + P_\lambda(\beta + \lambda_F)) + AF(C_F - C_N - G + P_\lambda(\beta + \lambda_F + \lambda_N))}{A^2 + 2AF + F(F - 4B)} + \frac{F(-2B(C_F + P_\lambda(\beta + \lambda_F)) + F(C_N - G - P_\lambda \lambda_N))}{A^2 + 2AF + F(F - 4B)} \tag{16}$$

$$P_F^{**} = \frac{AB(C_F + P_\lambda(\beta + \lambda_F)) - BF(C_F + 2C_N + 2G + P_\lambda(\beta + \lambda_G - 2\lambda_N))}{A^2 + 2AF + F(F - 4B)} + \frac{AF(C_N - P_\lambda \lambda_N) + F^2(C_N - P_\lambda \lambda_N)}{A^2 + 2AF + F(F - 4B)} \tag{17}$$

Placing Equations (16) and (17) into Equations (3) and (4), generates the optimal production volume:

$$Q_N^{**} = \frac{A^2(C_F + G + P_\lambda(2 + \lambda_F)) - BF(C_F - 2C_N + 2G + P_\lambda(\beta + \lambda_F + 2\lambda_N))}{G(A^2 + 2AF + F(F - 4B))} + \frac{A(-B(C_F + P_\lambda(\beta + \lambda_F)) + F(C_F - 2C_N + G - P_\lambda \lambda_N(\beta + \lambda_F + 2\lambda_N))}{G(A^2 + 2AF + F(F - 4B))} \tag{18}$$

$$Q_F^{**} = \frac{A^2(C_N - P_\lambda \lambda_N) - A(-FC_N + FP_\lambda \lambda_N + B(2C_F + C_N + G + 2P_\lambda(\beta + \lambda_F) - P_\lambda \lambda_N))}{G(A^2 + 2AF + F(F - 4B))} + \frac{B(2B(C_F + P_\lambda(\beta + \lambda_F)) + F(-C_N + G + P_\lambda \lambda_N)}{G(A^2 + 2AF + F(F - 4B))} \tag{19}$$

**Proposition 6.** *There is a credit price $P_\lambda^{(2)}$. When $0 < P_\lambda < P_\lambda^{(2)}$, the total revenue decreases as the credit price increases. When $P_\lambda^{(2)} < P_\lambda$, the total revenue increases as the credit price increases.*

The proof is the same as with Proposition 5.

Proposition 6 Description: When the credit price is lower than the threshold, the cost of offsetting the negative credits of CAFC is low, and the sales price of fuel vehicles is still relatively low. At this time, consumer anticipation of regret for buying fuel vehicles are lower compared to new energy vehicles. Therefore, automakers produce more fuel vehicles.

As the price of credits increases, the cost of offsetting negative CAFC credits increases. This decreases the total profit. When the point price exceeds the threshold, the profit from the sale of NEV positive points becomes an important component of the automaker's profit. At this time, the automaker produces more new energy vehicles, generating a higher income in the point transaction. This increases the profit.

**Proposition 7.** *The increase in the CAFC credit coefficient and the NEV credit coefficient both lead to a decrease in the production of fuel vehicles. The magnitude of this reduction increases as the expected regret anticipation increases. This increases production, and this increase becomes larger as the anticipation of regret increases.*

The proof is the same as for Proposition 1 and Proposition 2.

## 6. Numerical Study

To apply the theoretical results obtained in the previous sections of this paper, this section evaluates a set of numerical studies, drawing from similar numerical studies in the literature, such as Tang et al. [10] and Lu et al. [29].

Assume that $\gamma = 0.5$, $\theta = 0.75$, $\lambda_F = 0.358$, $\lambda_N = 3.8$, $C_F = 0.125$, $C_N = 0.4$, $\beta = 0.12$.

There are three parts to the numerical simulation. For the first part, this section focuses on exploring the influence of consumer anticipated regret on the optimal production volume, and the corresponding optimal income of new energy vehicles and fuel vehicles under different credit prices. For the second part, this section analyzes the influence of the credit price on the benefits of all parties under different decision-making modes, and analyzes the influence of the credit price on the credit income under different modes. For the third part, this section analyzes the automaker's profit under optimal production decision from both supply and demand perspectives. This approach considers the dual-credit policy and anticipated regret.

### 6.1. The Influence of Regret Anticipation on Optimal Production Quantity and Optimal Profit

Figure 1a shows that as the regret sensitivity coefficient increases, the optimal production volume of new energy vehicles decreases. Since the sales price of new energy vehicles is higher, there is an increase in consumers' anticipated regret (the regret sensitivity coefficient will increase). This may increase the anxiety of consumers to buy new energy vehicles, resulting in a decrease in the sales of new energy vehicles and a decrease in the optimal production quantity.

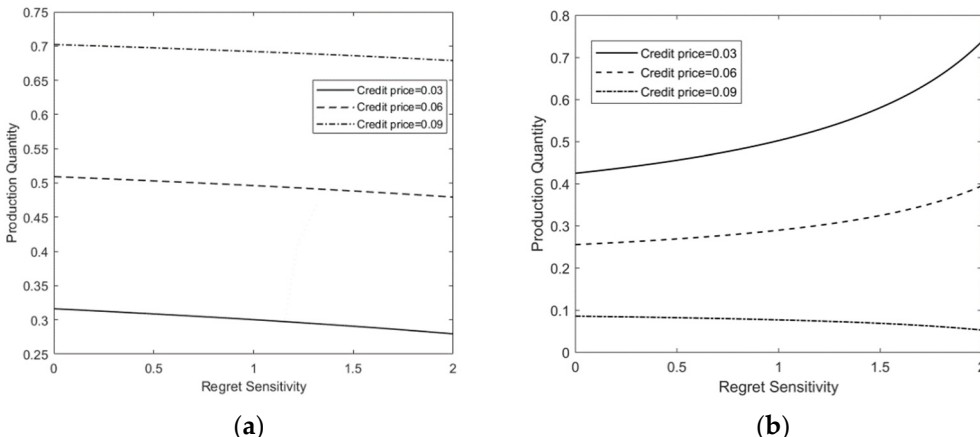

**Figure 1.** The impact of regret anticipation on vehicle production under independent decision-making. (**a**) The impact of regret anticipation on NEV production; (**b**) The impact of regret anticipation on FV production.

Figure 1b shows that when the credit price is low (the credit price is 0.03 or 0.06), the optimal production volume of fuel vehicles increases. When the credit price is high (the

credit price is 0.09), there is a decrease in the optimal production volume of fuel vehicles. When the price of credits is low, there is a lower cost associated with offsetting the negative credits of CAFC; fewer positive credits of NEV are required and the sales price of fuel vehicles remains low. Therefore, consumers are more willing to choose the fuel vehicle when there is stronger anticipation of regret. This increases the production quantity of the fuel vehicle production department. When the credit price is high, the cost of buying the same amount of NEV positive credits increases. This increases the sales price of fuel vehicles and strengthens consumer anticipation of regret. This leads to a decline in fuel car sales and production numbers.

Figure 2 shows that as the regret sensitivity coefficient increases, the total profit of the new energy vehicle production department decreases. The total profit of the fuel vehicle production department first decreases and then increases. When the automaker and the two departments see an increase in consumers' anticipated regret, the automaker and the two departments lower the sales price of the product to lower the anticipated regret, decreasing the total profit of the automaker's two departments. When the anticipation of regret is high enough, consumers favor products that have lower selling prices, increasing the total profit of the fuel vehicle production department.

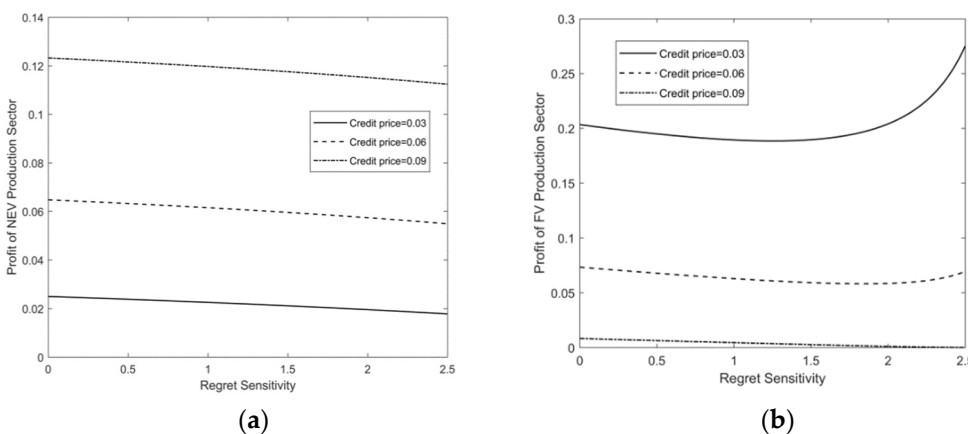

**Figure 2.** Influence of regret sensitivity coefficient on profits of the two departments under independent decision-making. (**a**) The impact of regret anticipation on NEV profits; (**b**) The impact of regret anticipation on FV profits.

*6.2. The Influence of Point Price on the Benefits of All Parties and Point Income under Different Decision-Making Modes*

Figure 3 shows the following.

When the credit price is lower than the threshold, the total profit is higher in the independent decision-making mode compared to the centralized decision-making mode. When the credit price is higher than the threshold, the total profit is lower in the independent decision-making mode compared to the centralized decision-making mode. When the price of credits is lower than the threshold, the cost of offsetting the negative credits of CAFC is lower. At this time, fuel vehicles continue to have a price advantage. Influenced by anticipation of regret, consumers are incentivized to buy fuel vehicles, increasing the profit generated by producing fuel vehicles. Under centralized decision-making, the negative credits of CAFC generated by producing fuel vehicles can be offset by the positive credits of NEV generated by producing new energy vehicles. Therefore, automakers balance the production quantities of the two types of vehicles to balance the credits. However, when the price of credits is lower, the profit generated by producing new energy vehicles is lower. As such, the profit is lower under centralized decision-making compared to under independent decision-making. When the credit price is higher than the threshold, as the credit price increases, there is an increase in the income generated from the point transaction of producing new energy vehicles. This gradually accelerates the increase in

the total profit. Therefore, the automaker is incentivized to produce new energy vehicles. The credit transaction thereby generates higher returns.

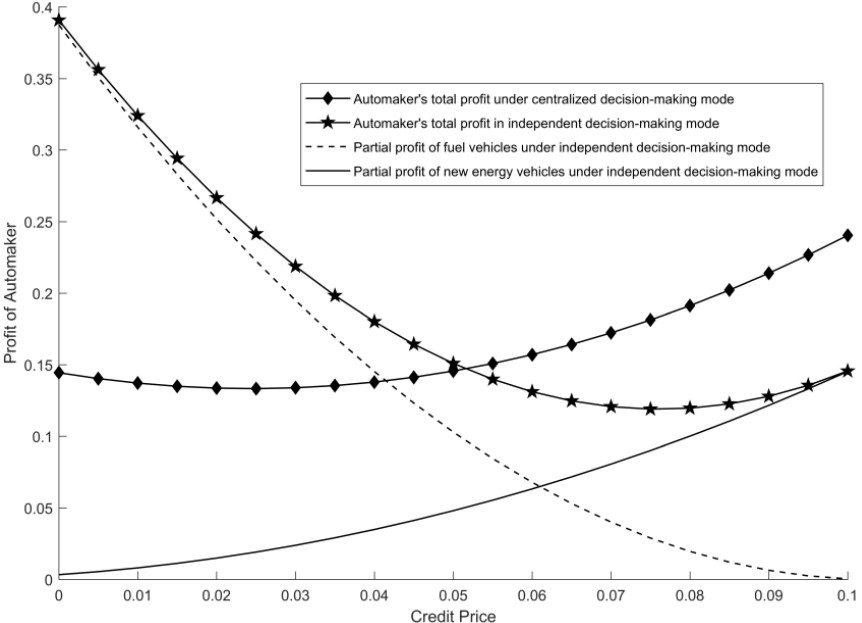

**Figure 3.** Comparison of the impact of credits price on profit.

In the independent decision-making model, the profit of the new energy vehicle production department increases, and the profit of the fuel vehicle production department decreases. As the price of credits increases, the income from selling NEV positive credits increases, and the cost of purchasing NEV positive credits increases. Therefore, the production of new energy vehicles increases and the profit from producing fuel vehicles decreases.

As the integral price increases, the total profit of the automaker in the two decision-making modes first decreases and then increases. From a market perspective, the government should reasonably adjust the price of credits to ensure the overall development of the auto industry. At the same time, automakers can increase the benefits of credit trading by reducing the average fuel consumption of fuel, and improving the battery life of new energy vehicles.

Figure 4 shows that, as the price of credits increases, there is an increase in the income from credit transactions in both decision modes. When the credit price is lower than the threshold, the benefits obtained through credit trading under independent decision-making are higher than centralized decision-making. When the price of credits is low, the direct benefits brought by the sale of fuel vehicles are higher than the benefits generated by the sale of new energy vehicles. The cost of offsetting negative CAFC credits remains low, so automakers under centralized decision-making are incentivized to produce fuel vehicles. This generates more negative CAFC credits compared to under independent decision-making. As such, the benefits of credit trading under centralized decision-making are lower than under independent decision-making. When the price of credits exceeds the threshold, selling NEV positive credits is an important part of the automaker's profit, and the cost of offsetting negative CAFC credits is higher. Under centralized decision-making, to obtain more NEV positive points, automakers increase the production quantity of new energy vehicles, generating more NEV positive credits compared to independent decision-making. This, in turn, generates higher returns in credit transactions.

*6.3. The Impact of Supply and Demand on Automakers' Earnings under Optimal Production Decisions*

Figure 5a shows that as the price of credits increases, the profit associated with producing fuel vehicles decreases and the profit from producing new energy vehicles increases.

Figure 5b shows that as the NEV credit proportional coefficient increases, the income from producing new energy vehicles increases, and the income of producing fuel vehicles decreases. As the regret sensitivity coefficient increases, the income of producing new energy vehicles decreases, and the increase in the income of the automaker increases the proportional coefficient of NEV credits. This balances the interests of the two departments of the automaker. At this stage, consumers' anticipated regret does not support the development of new energy vehicles. Automakers should reduce the impact of anticipated regret on new energy vehicles through outreach, R&D, and promotion. At the same time, the government should identify consumer sensitivity to regret in a timely manner, and balance the interests of different subjects by adjusting the intensity of policies, allowing the smooth transformation of automakers.

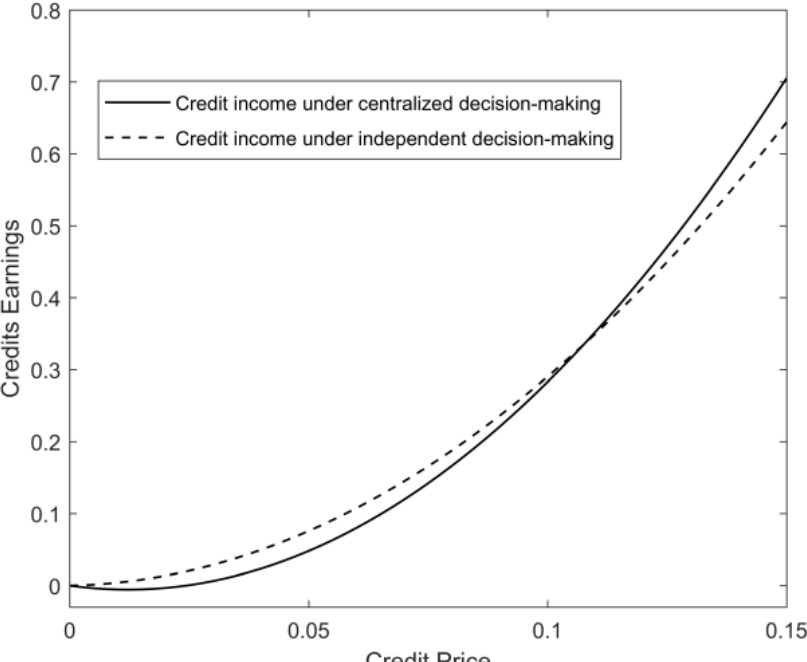

**Figure 4.** Comparison of the impact of credit prices on point transactions.

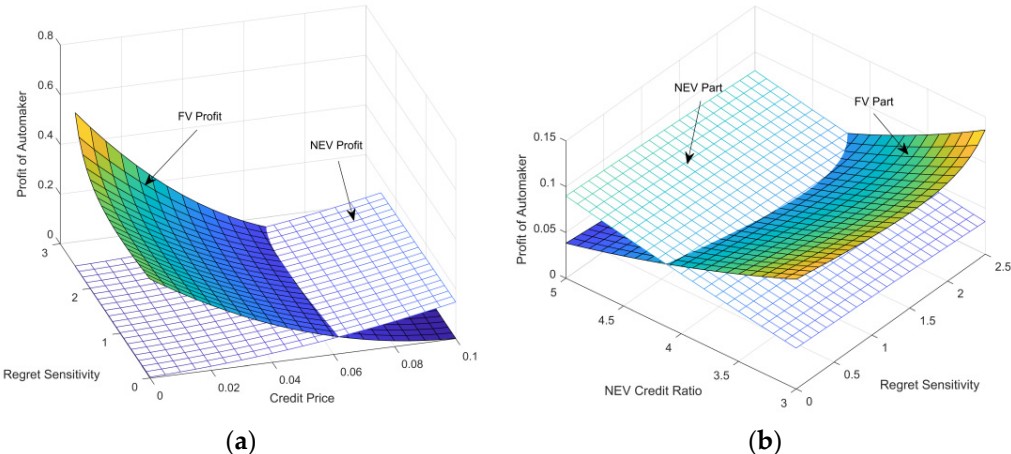

**Figure 5.** Comparison of the effects of regret anticipation and dual-credit parameters on the profits of two departments of the automaker under the independent decision-making mode. (**a**) The impact of regret anticipation and credit price on the profits of two departments; (**b**) The impact of regret anticipation and NEV credit ratio on the profits of two departments.

### 7. Summary and Outlook

Based on a consideration of the dual-credit policy, combined with regret anticipation psychology, this research focuses on an automaker that produces both fuel vehicles and new energy vehicles, and constructs a consumer surplus model and an automaker income model. The key findings were as follows:

Firstly, increases in the CAFC credit coefficient and the NEV credit coefficient lead to a decrease in the production of fuel vehicles. This reduction increases as the anticipation of regret increases. The increase in production volume caused by the increase in the NEV credit coefficient increases as the anticipation of regret increases. In the independent decision-making mode, the increase in the production volume caused by the increase in the CAFC credit coefficient weakens as the expected regret increases. In the centralized decision-making mode, the increase in the production volume caused by the increase in the CAFC credit coefficient also increases with the increased anticipation of regret.

Secondly, with the increase in the price of credits, the credits income generated from producing new energy vehicles increases, and the optimal sales price of new energy vehicles decreases. This, in turn, reduces the expected negative utility of consumers' regret when purchasing new energy vehicles, and improves the efficiency of new energy vehicles. In contrast, the negative credits paid to produce fuel vehicles offset the increase in cost, and the best-selling price of fuel vehicles increases. This increases the expected disutility of consumers buying and then regretting the car purchase, reducing the production quantity of fuel vehicles. At this stage, there remain many defects in new energy vehicles. For example, the battery life is significantly affected by the season, and the battery safety is low. When simply considering the potential for regret, fuel vehicles are more useful to consumers. Therefore, the government should reasonably regulate the price of points to prevent fuel vehicles from being squeezed out of the market, while promoting the development of new energy vehicles. Automakers should continue to invest in research and development; break through technical bottlenecks to improve consumer utility when considering purchasing new energy vehicles; establish advantages in price, performance, environmental attributes, and other aspects; and create a healthy industry ecology that does not require external policy promotion.

Thirdly, the increase in the NEV credit ratio requirement simultaneously increases the prices of both products. The price of fuel vehicles increases by a larger margin; the expected negative utility created from the regret of purchasing a fuel vehicle increases; and consumers regret the expected negative utility of purchasing new energy vehicles. This leads to a decrease in the production of fuel vehicles and an increase in the production of new energy vehicles. Increasing the percentage of credits means that more stringent production requirements are proposed for automakers. Automakers should meet the dual-credit policy by enhancing the battery life of new energy vehicles and increasing the number of positive credits created by new energy vehicles per production unit. Therefore, the government should increase its oversight of the market, urge automakers to invest in research and development, and allow the price of automobile products to fluctuate within a reasonable range.

Fourthly, under the trend of an increasing regret sensitivity coefficient, the optimal production volume of new energy vehicles decreases; when the credit price is low, the optimal production volume of fuel vehicles increases; and when the credit price is high, the optimal production volume of fuel vehicles decreases. With the increase of the regret sensitivity coefficient for new energy vehicles, the total profit of the production department decreases, and the total profit of the fuel vehicle production department first decreases and then increases. Therefore, there is uncertainty in the influence of consumers' anticipated regret on the automaker's two sectors. To this end, the government should work to understand the psychological dynamics of consumers accurately and quickly, and work to reduce consumer regret anticipation, and anxiety when purchasing new energy vehicles. One way to do this is by enriching the way of publicity. At the same time, automakers can

alleviate consumers' anticipated regrets by disclosing product information and optimizing the after-sales guarantee system.

This paper focused on the dual-credit policy and anticipation of consumer regret when the credit price is determined. In reality, the price of credits changes with the supply and demand of NEV credits, and does not consider the R&D investment of automakers. Since a static game model is adopted in this paper, it is difficult to describe some rules under the dual-credit policy. For example, CAFC positive credits can be carried over to the next cycle according to the proportion. Therefore, in the future research, the author will consider adopting a dynamic game that fits the reality, and explore the coping strategies of automakers under the dual-credit policy. At the same time, at this stage, the subsidy policy remains in place, and automakers are affected by multiple policies. Follow-up research should focus on these and other related issues.

**Author Contributions:** Conceptualization, L.S.; methodology, L.S.; software, Y.W.; validation, L.S. and S.L.; formal analysis, Y.W.; investigation, L.S.; resources, L.S.; data curation, S.L.; writing—original draft preparation, Y.W.; writing—review and editing, L.S.; visualization, Y.W.; supervision, L.S.; project administration, L.S.; funding acquisition, L.S. All authors have read and agreed to the published version of the manuscript.

**Funding:** This study was supported by the National Natural Science Foundation of China (Nos.71874071 and 71473107), the Ministry of Education in China Youth Fund Project of Humanities and Social Sciences (No.16YJCZH153), and Postgraduate Research & Practice Innovation Program of Jiangsu Province (No.SJCX21_1669).

**Institutional Review Board Statement:** Not applicable.

**Informed Consent Statement:** Not applicable.

**Data Availability Statement:** The dual-credit policy rules and related information are from the Ministry of Industry and Information Technology of the People's Republic of China. The specific information can be found here: https://www.miit.gov.cn/zwgk/zcwj/flfg/art/2020/art_2337a6d7ca894c5c8e8483cf9400ecdd.html (accessed on 8 July 2020).

**Conflicts of Interest:** The authors declare no conflict of interest.

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
