# Peer review of "Production Decisions of Automakers Considering the Impact of Anticipated Regret under the Dual-Credit Policy"

_sustainability, doi:10.3390/su14116598_

Round 1
Reviewer 1 Report
Please could you add a short paragraph to outline how the assumptions made in order to develop and employ the model stack up againt the reality of the system from an industrial perspective i.e. how useful is the approach to industry given the assumptons you have made to develop the model.
Outline some of the limitations of this approach too
Author Response
Reviewer 1
- Please could you add a short paragraph to outline how the assumptions made in order to develop and employ the model stack up againt the reality of the system from an industrial perspective i.e. how useful is the approach to industry given the assumptions you have made to develop the model.
★Answer:
Thanks for your these comments and suggestions. In real life, it is not uncommon that consumers have anticipated regret in the context of car purchases. The example, of Tesla’s seven-day plan for no-reason return a few years ago indirectly proved consumers’ regret before and after the decision to buy a car. In China, there are cases of consumers’ regretting purchasing cars, too. For example, on April 19, 2021, the "roof protection" incident of Tesla owners in Shanghai, China, , shows that the regret experience is bad for consumers. And when the concept of regret anticipation was put forward, the author explained that the emergence of consumers’ psychology on regret anticipation is due to t their many compunctious experiences, e which is common. Therefore, this paper only briefly explains the connection between this paper and practice in the introduction part. At the same time, in the model hypothesis, it is difficult to completely restore and describe the irrational behavior of consumers with a mathematical model. Therefore, the hypothesis in this paper is more to limit the performance of consumers' anticipated regret, and to make the model match the actual situation as perfectly as possible. Therefore, the assumption is mostly carried out in the anticipated regret.
- Outline some of the limitations of this approach too.
★Answer:
Thanks for your these comments and suggestions. Corresponding content has been added in the last paragraph, as follows: Since a static game model is adopted in this paper, it is difficult to describe some rules under the duel-credit policy. For example, CAFC positive credits can be carried over to the next cycle according to the proportion. Therefore, in the future research, the author will consider adopting a dynamic game that fits the reality, and explore the coping strategies of automakers under the duel-credit policy.
Reviewer 2 Report
Manuscript entitled "Production decisions of automakers considering the impact of anticipated regret under the dual-credit policy" of interest to a highly ranked journal like "Sustainability". Specifically, the authors propose appropriate scientifically sound solutions based on two aspects: production decisions and the anticipated regret of consumers.
Overall, the topic of this study is relevant, and the manuscript was well organized and written. This research focuses on an automaker that produces both fuel vehicles and new energy vehicles, and constructs a consumer surplus model and an automaker income model.
I hope that next suggestions can help to improve the manuscript.
- Add a conceptual model or study framework for better perception of article.
- More clear limitations (e.g. product quality attributes) should be included in conclusion part.
- Specify the authors’ contribution.
Author Response
Reviewer 2
Manuscript entitled "Production decisions of automakers considering the impact of anticipated regret under the dual-credit policy" of interest to a highly ranked journal like "Sustainability". Specifically, the authors propose appropriate scientifically sound solutions based on two aspects: production decisions and the anticipated regret of consumers.
Overall, the topic of this study is relevant, and the manuscript was well organized and written. This research focuses on an automaker that produces both fuel vehicles and new energy vehicles, and constructs a consumer surplus model and an automaker income model.
- Add a conceptual model or study framework for better perception of article.
★Answer:
Thanks for your these comments and suggestions. This part of the content has been added in the newly added third section, as follows: the chapters of this paper are described as follows: the problems and the model assumptions are described in Section 4. Section 5 mainly conducts a theoretical analysis on the influence of duel-credit policy and the anticipated regret psychology on the production decision of the automaker. Section 6 verifies the theoretical conclusions obtained in Section 5 through numerical simulations, and attempts to find conclusions that are difficult to obtain through mathematical models Section 7 summarizes the results obtained in this paper and introduces the directions of author's future research.
- More clear limitations (e.g. product quality attributes) should be included in conclusion part.
★Answer:
Thanks for your these comments and suggestions. The quality attributes of the product are generally reflected in the prices of products. For example, the iPhone 13 Pro adds a function of 120Hz screen refresh rate, compared to the iPhone 13, so the price of iPhone 13 Pro is higher than that of the iPhone 13. For consumers, the price of a product is the information intuitively perceived more. In the same way, for car purchasing the product attributes of NEV and FV, including battery quality, fuel economy, etc., determine their prices with the different parts. However, consumers with regret expectations are more focused on the difference of product information between NEV and FV (which is embodied in prices in this article.), and this article only focuses on the production decisions of automakers, and explores the changes in the single quality attributes of the products, which is more in line with the R&D decisions of automakers. Therefore, this paper simply believes that consumers intuitively feel product attributes by observing products’ prices.
- Specify the authors’ contribution.
★Answer:
Thanks for your these comments and suggestions. Author contributions have been added in the newly added third section, as follows: Regarding the method and contribution, the duopoly game model is adopted in this paper. Although this article takes one automaker as the research object, the automaker has two departments (NEV production department and FV production department, details in the problem description ). Then this paper explores the influence of duel-credit policy and anticipated regret on production decision under centralized and independent decision-making. The contribution of this paper to the domain of NEV is the construction of the demand function based on the anticipated regret function. In previous papers, scholars have constructed demand functions based either on consumer surplus or on market demand. In this paper, the anticipated regret function is used to construct the demand function, which enriches the construction method of the demand function in the NEV automotive field. And in the existing literature, few scholars have introduced regret anticipation psychology in the field of NEV. This paper fills this gap.
Reviewer 3 Report
The article „Production decisions of automakers considering the impact of anticipated regret under the dual-credit policy” introduces the psychology of consumers' anticipated regret, analyzes the production decisions of automakers, and explores the optimal production decisions. The topic of the article is mainly related to economics sciences.
The title of the article corresponds to its content. The abstract and keywords also match the content of the article. The abstract outlines the contribution of the paper (the research design, the methods and procedures employed, the main outcomes and results). This part of the article includes key findings and is an appropriate length. However, keywords should not be written with capital letters.
The purpose of the article is clearly stated. The introduction is effective, clear and well organized. This section introduce and put into perspective what follows. But I suggest develop a background indicating a research gap and present what is novel and why it is significant. A gap in knowledge is not identified.
The scope and manner of using the literature on the subject are not objectionable. The cited references are current (mostly within the last 5 years).
The general structure of the reviewed article is correct. The order of the individual subsections and the content presented in them are also correct. The research methods used by the authors are appropriate. Unfortunately, the research method applied is not clearly stated. The description of the method should definitely be supplemented. There is also no description of the limitations in the studies carried out. The research period was not indicated either. In my opinion, a separate section should be created for the description of the research methodology.
I rate positively the publication content (placing the problem in the right context) and the formal correctness (formulas, algorithms). But the authors did not formulate research questions or hypotheses, which decreases the scientific value of the article.
Stages of description, analysis and interpretation are properly carried out. Illustrative material is logically related to the article. The figures and tables are appropriate and properly show the data.
Conclusion clearly presents the research findings and recommendations. The conclusions are supported by research results. The assumptions formulated for the purpose of work have been implemented. But the Conclusion section should be developed and indicate advantages, limitations and potential directions for further research. The article also measures the comparison or discussion of the results with the results of other researchers. The formulated conclusions are important, although the article does not directly bring new knowledge to science.
The language of the article corresponds to the correctness criteria used in scientific statements. The language is clear. But I do not feel an expert in assessing the language quality of the article. I suggest having the manuscript proof read and edited before submitting.
The bibliography has not been prepared in accordance with the publisher's guidelines. Some carelessness in the preparation of the list of references is evident. In addition, I do not think it is necessary to put an enumeration mark at the beginning of a paragraph. On pages 11-14, the Authors used enumeration characters, i. e. (1), (2) etc. at the beginning of the paragraphs. I suggest the Authors dispense with that.
The article meets the requirements of this type of publication. The research methods used may be of interest to other researchers and with appropriate modification can be used for further research. The article does not explicitly refer to the issue of the concept of sustainable development. That’s why the article only indirectly fits into the theme of Sustainability.
Author Response
Reviewer 3
- However, keywords should not be written with capital letters.
★Answer:
Thanks for your these comments and suggestions. Keywords have been modified, as follows: Keywords: new energy vehicle; dual-credit policy; anticipated regret; production decision
- But I suggest develop a background indicating a research gap and present what is novel and why it is significant. A gap in knowledge is not identified.
★Answer:
Thank you for your suggestion. In the newly added third section, this section has been added, as follows: To highlight the novelty more clearly, Table 1 in this section presents the main differences between the content of this paper and related research. Since the implementation of the duel-credit policy, the production decision-making of automakers has been the focus of scholars' research. Yu et al. (2021) explored the impact of the subsidy policy and the duel-credit policy on the production decisions of automakers when the two types of vehicles (NEV and FV) were in independent markets. For consumers, NEV and FV, however, are substitute products, so the relationship between NEV and FV tends to be more competitive in practice. Although Li et al. (2020) explored the optimal production decisions of automakers in a competitive market environment, they did not considered the psychological characteristics of consumers In a competitive environment, Tang et al. (2021) considered the impact of consumers' low-carbon preference on the production decisions of the automakers, but low-carbon preference only showed consumers' attention to the advantages of NEV, while ignoring consumers' attention to the advantages of FV , such as price advantage. The anticipated regret highlights the entanglement of consumers facing two competing products before making a purchase decision. In the introduction, this article explains why anticipation of regret precedes the decision to purchase cars. Therefore, this paper will investigate the influence of consumers' anticipated regret on the optimal decisions of production for the automaker under the duel-credit policy.
Table 1. Main differences between our works and existing research.
Articles |
Research object |
Market environmen |
Focus Point |
consumer psychology |
Lou et al. [7] |
ICEV |
independent |
R&D, Production |
low carbon preference |
Tang et al. [10] |
NEV, FV |
competitive |
Price, Production |
low carbon preference |
Lu et al. [29] |
NEV, FV |
competitive |
Price, Production, R&D |
low carbon preference |
Li et al. [9] |
NEV, FV |
independent |
Profit |
× |
Yang et al. [52] |
NEV, FV |
competitive |
Price Production, Profit |
× |
Li et al. [8] |
NEV, FV |
competitive |
Production |
× |
Ou et al. [53] |
NEV, FV |
competitive |
Production, R&D |
× |
Yu et al. [28] |
NEV ,FV |
independent |
Production, R&D |
× |
Cheng et al. [26] |
NEV, FV |
independent |
Production |
× |
Ma et al. [54] |
FV |
independent |
Production, R&D |
× |
- Unfortunately, the research method applied is not clearly stated. The description of the method should definitely be supplemented. There is also no description of the limitations in the studies carried out. The research period was not indicated either. In my opinion, a separate section should be created for the description of the research methodology.
★Answer:
Thanks for your these comments and suggestions. Corresponding content has been added to the newly added third section and last paragraph, as follows: Regarding the method and contribution, the duopoly game model is adopted in this paper. Although this article takes one automaker as the research object, the automaker has two departments (NEV production department and FV production department, details in the problem description ). Then this paper explores the influence of duel-credit policy and anticipated regret on production decision under centralized and independent decision-making. The contribution of this paper to the domain of NEV is the construction of the demand function based on the anticipated regret function. In previous papers, scholars have constructed demand functions based either on consumer surplus or on market demand. In this paper, the anticipated regret function is used to construct the demand function, which enriches the construction method of the demand function in the NEV automotive field. And in the existing literature, few scholars have introduced regret anticipation psychology in the field of NEV. This paper fills this gap. Corresponding content has been added in the last paragraph, as follows: Since a static game model is adopted in this paper, it is difficult to describe some rules under the duel-credit policy. For example, CAFC positive credits can be carried over to the next cycle according to the proportion. Therefore, in the future research, the author will consider adopting a dynamic game that fits the reality, and explore the coping strategies of automakers under the duel-credit policy.
- But the authors did not formulate research questions or hypotheses, which decreases the scientific value of the article.
★Answer:
Thanks for your these comments and suggestions. After the literature review, this paper raises three questions for the past research, and proposes the question to be studied in this paper (see the last two paragraphs of Section 2 for details).
- But the Conclusion section should be developed and indicate advantages, limitations and potential directions for further research. The article also measures the comparison or discussion of the results with the results of other researchers. The formulated conclusions are important, although the article does not directly bring new knowledge to science.
★Answer:
Thanks for your these comments and suggestions. Firstly, indicate advantages and comparison of the results with the results of other researchers are added in the newly added third section as follows: To highlight the novelty more clearly, Table 1 in this section presents the main differences between the content of this paper and related research. Since the implementation of the duel-credit policy, the production decision-making of automakers has been the focus of scholars' research. Yu et al. (2021) explored the impact of the subsidy policy and the duel-credit policy on the production decisions of automakers when the two types of vehicles (NEV and FV) were in independent markets. For consumers, NEV and FV, however, are substitute products, so the relationship between NEV and FV tends to be more competitive in practice. Although Li et al. (2020) explored the optimal production decisions of automakers in a competitive market environment, they did not considered the psychological characteristics of consumers In a competitive environment, Tang et al. (2021) considered the impact of consumers' low-carbon preference on the production decisions of the automakers, but low-carbon preference only showed consumers' attention to the advantages of NEV, while ignoring consumers' attention to the advantages of FV , such as price advantage. The anticipated regret highlights the entanglement of consumers facing two competing products before making a purchase decision. In the introduction, this article explains why anticipation of regret precedes the decision to purchase cars. Therefore, this paper will investigate the influence of consumers' anticipated regret on the optimal decisions of production for the automaker under the duel-credit policy.
Secondly, The limitations and potential directions are modified and reflected in the most paragraph, as follows: This paper focused on the dual-credit policy and anticipation of consumer regret when the credit price is determined. In reality, the price of credits changes with the supply and demand of NEV credits, and does not consider the R&D investment of automakers. Since a static game model is adopted in this paper, it is difficult to describe some rules under the duel-credit policy. For example, CAFC positive credits can be carried over to the next cycle according to the proportion. Therefore, in the future research, the author will consider adopting a dynamic game that fits the reality, and explore the coping strategies of automakers under the duel-credit policy. At the same time, at this stage, the subsidy policy remains in place, and automakers are affected by multiple policies. Follow-up research should focus on these issues.
- The bibliography has not been prepared in accordance with the publisher's guidelines. Some carelessness in the preparation of the list of references is evident.
★Answer:
Thank you for pointing out this weakness, and we apologize for our carelessness. These have been modified.
- In addition, I do not think it is necessary to put an enumeration mark at the beginning of a paragraph. On pages 11-14, the Authors used enumeration characters, i. e. (1), (2) etc. at the beginning of the paragraphs. I suggest the Authors dispense with that.
★Answer:
Thank you for your suggestion. These have been modified.
- The article does not explicitly refer to the issue of the concept of sustainable development. That’s why the article only indirectly fits into the theme of Sustainability.
★Answer:
Thanks for your these comments and suggestions. China's 14th Five-Year Plan clearly pointed out that as for sustainable development, the emissions of carbon dioxide per unit of GDP should be reduced by 18%, and that the development of NEVs is an effective way to achieve this goal. Hence, the first sentence of this article states that “with the increasing pressure of emission reduction, new energy vehicles have become the focus of many scholars and governments due to their excellent environmental effects”, thus the main part of this article comes soon.